# The Predictive Value of Low Skeletal Muscle Mass Assessed on Cross-Sectional Imaging for Anti-Cancer Drug Toxicity: A Systematic Review and Meta-Analysis

**DOI:** 10.3390/jcm9113780

**Published:** 2020-11-23

**Authors:** Laura F. J. Huiskamp, Najiba Chargi, Lot A. Devriese, Anne M. May, Alwin D. R. Huitema, Remco de Bree

**Affiliations:** 1Department of Head and Neck Surgical Oncology, UMC Utrecht Cancer Center, University Medical Center Utrecht, Utrecht University, Heidelberglaan 100, 3584 CX Utrecht, The Netherlands; L.F.J.Huiskamp@umcutrecht.nl (L.F.J.H.); N.Chargi-2@umcutrecht.nl (N.C.); 2Department of Medical Oncology, University Medical Center Utrecht, Utrecht University, Heidelberglaan 100, 3584 CX Utrecht, The Netherlands; L.A.Devriese@umcutrecht.nl; 3Department of Epidemiology, Julius Center for Health Sciences and Primary Care, University Medical Center Utrecht, Utrecht University, Heidelberglaan 100, 3584 CX Utrecht, The Netherlands; A.M.May@umcutrecht.nl; 4Department of Clinical Pharmacy, University Medical Center Utrecht, Utrecht University, Heidelberglaan 100, 3584 CX Utrecht, The Netherlands; A.D.R.Huitema-2@umcutrecht.nl; 5Department of Pharmacy & Pharmacology, Netherlands Cancer Institute, Plesmanlaan 121, 1066 CX Amsterdam, The Netherlands

**Keywords:** low skeletal muscle mass, cancer, meta-analysis, toxicity, anti-cancer drugs

## Abstract

Low skeletal muscle mass (LSMM) is increasingly recognized for its predictive value for adverse events in cancer patients. In specific, the predictive value of LSMM has been demonstrated for anti-cancer drug toxicity in a variety of cancer types and anti-cancer drugs. However, due to the limited sample size and study populations focused on a single cancer type, an overall predictive value of LSMM for anti-cancer drug toxicity remains unknown. Therefore, this review aims to provide a comprehensive overview of the predictive value of LSMM and perform a meta-analysis to analyse the overall effect. A systematic search was conducted of MEDLINE, Scopus, EMBASE, and Cochrane. Inclusion criteria were skeletal muscle mass (SMM) evaluated with computed tomography (CT) or magnetic resonance imaging (MRI), articles published in English, SMM studied in humans, SMM measurement normalized for height, and patients did not receive an intervention to treat or prevent LSMM. A meta-analysis was performed using a random-effects model and expressed in odds ratio (OR) with 95% confidence interval (CI). Heterogeneity was assessed using χ^2^ and I^2^ statistics. The search yielded 907 studies. 31 studies were included in the systematic review. Sample sizes ranged from 21 to 414 patients. The occurrence of LSMM ranged from 12.2% to 89.0%. The most frequently studied cancer types were oesophageal, renal, colorectal, breast, and head and neck cancer. Patients with LSMM had a higher risk of severe toxicity (OR 4.08; 95% CI 2.48–6.70; *p* < 0.001) and dose-limiting toxicity (OR 2.24; 95% CI 1.28–3.92; *p* < 0.001) compared to patients without LSMM. To conclude, the predictive value of LSMM for anti-cancer drug toxicity can be observed across cancer types. This information increases the need for further research into interventions that could treat LSMM as well as the possibility to adapt treatment regimens based on the presence of LSMM.

## 1. Introduction

There is a high prevalence of low skeletal muscle mass (LSMM), sometimes referred to as sarcopenia, in cancer patients. Moreover, in advanced stages of cancer, the majority of patients exhibit LSMM [1,2]. A large number of studies has been performed to investigate the predictive value of LSMM. Especially, the association between LSMM and survival has been thoroughly investigated [2,3,4]. This prognostic value of LSMM has been demonstrated in a variety of cancer types including lung [3], colorectal [5], breast [6], renal [7], and head and neck cancer [8]. LSMM has also been investigated as a predictive factor for adverse events such as chemotherapy toxicity, surgical complications, and radiotherapy toxicity [5,6,7,9,10].

There are several techniques for the measurement of skeletal muscle mass (SMM). This includes dual-energy X-ray absorptiometry (DXA), which uses x-rays that will reduce in energy based on the composition and thickness of the material that it passes through, and bioelectric impedance analysis (BIA), which measures body composition using an electrical current that experiences more resistance through adipose tissue as opposed to electrolyte-rich fluids [5,11]. The most commonly used technique utilizes computed tomography (CT) as it is part of routine care in the majority of cancer patients, and it has a proven high accuracy in measuring SMM [3,8,12]. Most studies quantify SMM using CT scans of the third lumbar (L3) vertebrae, although other levels have also been used. The cross-sectional area (CSA) of skeletal muscle mass is measured on a single cross-sectional image and normalized for height resulting in the skeletal muscle index (SMI). The SMI correlates strongly with total-body skeletal muscle mass [12,13]. Recently, magnetic resonance imaging (MRI) has been proven to have a strong correlation (r^2^ = 0.94, *p* < 0.01) with CT for the measurement of the CSA of SMM [14].

Although the predictive value of LSMM has been investigated frequently, the underlying mechanism is only hypothesized. There are theories about the underlying pathophysiology of LSMM such as the influence of age, intracellular oxidative stress, and genetic components [3,15]. In cancer patients, there is also a high possibility of developing cachexia which could also result in LSMM [3,15]. There are several theories for the mechanism by which LSMM influences toxicity. Some theorize that the altered ratio of fat-to-lean body mass can influence the pharmacokinetics of anti-cancer drugs [11]. Others theorize that LSMM is independently associated with frailty, which can result in a higher risk of adverse events [4,11,16]. The most commonly supported hypothesis is based on the influence of LSMM on drug distribution. The body consists of two major compartments, fat mass (FM) and lean body mass (LBM); drugs can be inclined to distribute towards one of these compartments. Patients with LSMM have a decreased LBM and, as muscle mass is the largest contributor to LBM, this may result in increased drug levels in the plasma and thereby a higher risk of toxicity [6,8,9,11].

Although there have been many studies devoted to the predictive value of LSMM for anti-cancer drug toxicity, these studies have several limitations, such as small sample sizes. Additionally, the majority of studies focus on a single cancer type or disease stage which limits its ability to draw conclusions for a large population of cancer patients [4,5,7]. To conclude whether this predictive value of LSMM is present across cancer types and treatments, studies have to be performed in a larger and wider population.

This systematic review aims to provide a comprehensive overview of the literature and data regarding the predictive value of LSMM for anti-cancer drug toxicity and analyse the overall effect in a meta-analysis. Specifically, this review will investigate whether this predictive value is universal across cancer types. Additionally, this review will study if there is a relationship between drug distribution and the predictive value of LSMM for anti-cancer drug toxicity.

## 2. Methods

### 2.1. Search Strategy

The systematic review was conducted according to the Preferred Reporting Items for Systematic Reviews and Meta-Analyses (PRISMA) standards [17]. A systematic search was performed in four electronic databases, which are MEDLINE, EMBASE, Cochrane, and Scopus, from inception through 17 February 2020. The search terms included toxicity, sarcopenia, chemotherapy, cancer, and synonyms for each of these terms detailed in Appendix A. The references of each included article were also screened to identify additional records.

### 2.2. Study Selection

The studies obtained from the systematic search were assessed by screening titles and abstracts, by a single researcher (L.F.J.H.) Subsequently, the potentially included articles were assessed using the full text. Studies were included in the analysis when they met the following inclusion criteria: (1) examine the association of LSMM and anti-cancer drug toxicity, (2) evaluate skeletal muscle mass by measuring cross-sectional area on CT or MRI, (3) are published in English, and (4) describe studies in humans only. Studies were excluded from the analysis when they met the following exclusion criteria: (1) do not normalize SMM for height; (2) are a systematic review, conference paper, or study protocol; or (3) only describe an intervention and its effects on SMM or toxicity.

### 2.3. Data Extraction

The data were extracted and collected from each included study. This consisted of (1) author and publication year, (2) population size and cancer type, (3) occurrence and definition of low SMM, (4) technique used for the evaluation of SMM (such as scan type, software for image analysis, and vertebrae level analysed), (5) treatment specifications (anti-cancer drug, curative or palliative intent, primary or adjuvant, and combination with radiotherapy), (6) time between scan and treatment, (7) measure and occurrence of toxicity. Only published data was included.

### 2.4. Assessment of Risk of Bias

The risk of bias was assessed using the Quality In Prognosis Studies (QUIPS) tool [18]. The QUIPS tool assesses the risk of bias based on six domains each with multiple sub-domains. Each sub-domain is rated with “yes”, “no”, “partial”, or “unsure” after which each domain is rated low, moderate, or high based on the ratings of the sub-domains. The six domains are (1) study participation, (2) study attrition, (3) prognostic factor measurement, (4) outcome measurement, (5) study confounding, and (6) statistical analysis and reporting [18]. A study was scored as low risk of bias when at least four domains were rated as low, and a maximum of two domains was rated moderate (of which prognostic factor measurement and outcome measurement must be rated low), with no domains rated as high. A study was scored as high risk of bias if more than two domains were rated high, or four domains were rated moderate. All remaining studies were scored as a moderate risk of bias.

### 2.5. Data Analysis

A meta-analysis was performed using Review Manager (Revman v5.3, The Nordic Cochrane Collaboration, Copenhagen, Denmark, 2014). A random-effects model was used because of the assumed heterogeneity between the studies. Studies were excluded from the meta-analysis if (1) there was insufficient data to calculate an odds ratio (OR); (2) LSMM was not defined with a cut-off value, and SMI was instead used as a continuous variable; or (3) the endpoint for toxicity did not match any other studies, hampering combination with other studies for meta-analysis.

The results were visualized using forest plots expressed in OR with 95% confidence interval (CI). The results were stratified for toxicity definition, namely, toxicity ≥ grade 3 according to Common Terminology Criteria for Adverse Events (CTCAE) and dose-limiting toxicity (DLT). Further stratification was based on cut-off values, measurement technique, and vertebrae level analysed. Heterogeneity was assessed with the χ^2^ and I^2^ statistic tests. I^2^ values between 25% and 50% were considered to demonstrate low heterogeneity, 50% to 75% demonstrates moderate heterogeneity, and >75% was considered to demonstrate high heterogeneity. Subgroup analysis was performed for any monotherapy which was used in the populations of more than one study. *p*-values < 0.05 were considered statistically significant.

## 3. Results

### 3.1. Search Results

The search yielded 906 hits. One additional study was included after the screening of all included articles reference lists. After the removal of 357 duplicates, the titles and abstracts of 550 studies were screened. The screening of abstracts and titles yielded 52 studies for full-text screening. After the full-text screening, 31 met all inclusion criteria and were included in this review [5,6,7,8,16,19,20,21,22,23,24,25,26,27,28,29,30,31,32,33,34,35,36,37,38,39,40,41,42,43,44]. The selection process with exclusion reasons is shown in Figure 1. A total of 19 studies were included in the meta-analysis. Studies were excluded from the meta-analysis because the study did not include sufficient data to calculate odds ratios (*n* = 6) [19,26,30,36,41,43], did not dichotomize LSMM (*n* = 4) [32,33,38,40], or featured a toxicity endpoint that did not match with any other studies (*n* = 2) [6,42].

### 3.2. Study Characteristics

Table 1 shows the characteristics of the included studies. Samples sizes ranged from 21 to 414 patients with a total sample size of 2918 patients. The study populations existed of patients with a variety of cancer types. The most frequent were oesophageal, renal, colorectal, breast, and head and neck cancer. The occurrence of LSMM ranged from 12.2% to 89.0%. The endpoint used to measure toxicity varied between studies. Most studies used DLT, defined as toxicity leading to dose reduction, treatment delay, or treatment discontinuation. Another common measurement of toxicity was according to the CTCAE grading system. The occurrence of toxicity ranged from 21.8% to 77.4%. Table A1 shows additional information regarding the treatment specificities of the included studies, such as treatment intent, primary or adjuvant treatment, and the addition of radiotherapy.

### 3.3. Skeletal Muscle Mass Assessment

There were several differences between the studies in the method used to measure SMM. All included studies used CT to evaluate SMM with one study also using MRI [29]. However, there was a difference in the selected vertebrae used for the SMM assessment as shown in Table 1. Most studies used lumbar level 3 (L3); other vertebrae that were used were cervical level 3 (C3) and thoracic level 4 (T4). Table A1 shows other differences between studies such as the time between CT and treatment start, as well as the software used to measure SMM.

The included studies also used different cut-off values for LSMM; this can be seen in Table 1. Most studies used cut-off values cited from previous articles. The most commonly used cut-off values were established by Prado et al., 2008 [45] (<52.4 cm^2^/m^2^ for men and <38.5 cm^2^/m^2^ for women), followed by Martin et al., 2013 [46] (<43 cm^2^/m^2^ for men if body mass index (BMI) ≤24.9 kg/m^2^ or <53 cm^2^/m^2^ for men if BMI > 25 kg/m^2^ and <41 cm^2^/m^2^ for women), Fujiwara et al., 2015 [47] (≤36.2 cm^2^/m^2^ for men and ≤29.6 cm^2^/m^2^ for women), and Caan et al., 2018 [48] (<40 cm^2^/m^2^). It is noteworthy that five studies cited the cut-off values of Prado et al. [45] but used other cut-off values in their analysis than those published by Prado et al. [5,7,20,41,43]. Four studies did not use cut-off values for LSMM and instead used continuous SMI during analysis [32,33,38,40].

### 3.4. Study Quality Assessment

The results of the QUIPS assessment of all included studies are summarized in Figure 2. Out of the 31 included studies, seven studies had a low risk of bias [6,16,19,24,25,34,38], 16 studies had a moderate risk of bias [5,7,8,21,22,23,26,28,29,30,31,32,35,37,43,44], and eight had a high risk of bias [20,27,33,36,39,40,41,42]. The domains study participation, study confounding, and statistical analysis and reporting were most frequently assessed as having a high risk of bias. Whereas the domains study attrition, prognostic factor measurement, and outcome measurement were most frequently assessed as having a low risk of bias.

### 3.5. Association between LSMM and Toxicity

Figure 3A shows the forest plot for the OR of 13 studies that used DLT as the measure of toxicity. Kurk et al., 2019 [31] performed two separate analyses in the same patient population receiving sequential treatments, 232 patients treated with Capox-B and 182 patients treated with Cap-B. These results were entered into the forest plot separately. Patients with LSMM had a significantly higher risk for DLT compared to patients without LSMM (OR 2.24; 95% CI 1.28–3.92, *p* < 0.001). Heterogeneity across studies was high (χ^2^ = 60.97 and I^2^ = 79%). Figure 3B,C show a selection of the 13 studies that used DLT as an endpoint. To create an analysis with less heterogeneity, studies were matched together based on identical cut-off values, measurement techniques, and vertebrae level analysed. The studies included in Figure 3B all used the cut-off values established by Martin et al., 2013 [46] and measured SMM at L3 using CT.

There was no association between LSMM and DLT (OR 1.98; 95% CI 0.76–5.22, *p* = 0.16). Heterogeneity across studies was high (χ^2^ = 24.48 and I^2^ = 84%). The studies included in Figure 3C all used cut-off values established by Prado et al., 2008 [45] as well as the same measurement technique at L3 using CT. There was no associated between LSMM and DLT (OR 1.87; 95% CI 0.32–10.93, *p* = 0.49). Heterogeneity across studies was high (χ^2^ = 60.97 and I^2^ = 79%).

Figure 4A shows the forest plot for the OR of 6 studies that used toxicity ≥ grade 3 according to the CTCAE as the measure for toxicity. Patients with LSMM had a significantly higher risk of ≥grade 3 toxicity compared to patients without LSMM (OR 4.08; 95% CI 2.48–6.70; *p* < 0.01). Heterogeneity across studies was low (χ^2^ of 1.14 and I^2^ of 0%). Figure 4B shows the forest plot for the OR of 3 studies that besides using the same toxicity description also used the same cut-off value, namely that established by Martin et al., 2013 [46], as well as the same measurement technique on CT at the L3 vertebrae. Patients with LSMM had a significantly higher risk of ≥grade 3 toxicity compared to patients without LSMM (OR 3.81; 95% CI 2.07-6.98; p <0.001). Heterogeneity across studies was low (χ2 of 0.13 and I2 of 0%).

Of the 31 studies included in this review, 19 were included in the meta-analysis. Six studies were excluded because there was not sufficient statistical data published to determine an OR [19,26,30,36,41,43]. Of these six, five concluded that there was no association between SMI and toxicity [26,30,36,41,43], and one concluded that a lower SMI was related to a higher risk of toxicity [19]. Four studies were excluded because they did not dichotomize SMI and instead performed the analysis with SMI as a continuous variable [32,33,38,40]. Of these four, one concluded no association [32], and three concluded that low SMI was related to increased toxicity occurrence [33,38,40]. Two studies were excluded from the meta-analysis because the toxicity endpoint did not match any of the other studies [6,42]. Of these two, one showed a negative association between toxicity occurrence and SMI [6], and one showed no association [42]. Of the seven studies excluded that demonstrated no association between sarcopenia, several provided a theory as to why this association was not demonstrated. Some of these studies hypothesized that the distribution of the anti-cancer drug investigated was not influenced by LSMM, because of the hydrophilic characteristics of the drug or because of the route of administration [26,30,36]. Other studies mentioned the variety of cut-off values used for LSMM, which originated in populations that differ from the investigated population and can be observed in the varying prevalence of LSMM between studies [32,41]. Although these studies did not find an association between LSMM and toxicity, some did observe other associations related to LSMM and toxicity. These associations include the association between sarcopenic obesity and toxicity [26]; muscle quality and toxicity [30]; muscle loss during treatment and toxicity [43]; and LSMM and survival [42].

### 3.6. Subgroup Analysis

The studies that investigated the influence of a monotherapy were used for a subgroup analysis. Three different drugs used in monotherapy were the topic of more than one study. Figure 5A shows the forest plot for the OR of toxicity in LSMM and non-LSMM patients treated with cisplatin or carboplatin. Cisplatin and carboplatin have an apparent volume of distribution which approximately equals total body water (40–60 L) [49,50]. These drugs were used as monotherapy in two studies [8,28] and showed an association between LSMM and toxicity (OR 3.06; 95% CI 1.45–6.44, *p* = 0.003) with moderate heterogeneity (χ^2^ = 1.98; I^2^ = 49%). Figure 5B shows the forest plot for LSMM and non-LSMM patients treated with sorafenib as a monotherapy, which has an apparent volume of distribution of 213 L [51]. These two studies [20,37] demonstrated an association between LSMM and toxicity (OR 5.60; 95% CI 2.01–15.59; *p* = 0.001) with low heterogeneity (χ^2^ = 0.33; I^2^ = 0%). Figure 5C shows the forest plot for LSMM and non-LSMM patients treated with sunitinib as a monotherapy, which has an apparent volume of distribution of 2230 L [52]. These two studies [7,23] showed no association between LSMM and toxicity (OR 1.27; 95% CI 0.08–20.94; *p* = 0.87) with high heterogeneity (χ^2^ = 5.62; I^2^ = 82%).

## 4. Discussion

In this review, 31 studies were evaluated, of which 19 were used in the meta-analysis. The meta-analysis showed that LSMM has predictive value for toxicity (DLT OR = 2.24 and ≥grade 3 toxicity OR = 4.08). Heterogeneity across studies using DLT as the outcome was very high, which can be explained by the differences in the definition of DLT. The general definition of DLT is any toxicity leading to dose reduction, treatment delay, or discontinuation. However, studies differed in the level of detail of this definition, for example, some studies included any dose reduction [19,24], others applied a minimum of 50% reduction [8,20], some studies also included toxicity leading to hospitalization [39], some studies included any treatment delay [19,24], and others included delays over 4 days [8] or 7 days [28]. Even when creating subgroups by matching cut-off values, measurement techniques, and vertebrae level analysed, the heterogeneity remained high because of the difference in the definition of DLT and could not provide accurate evidence of the association between LSMM and DLT. The results presented in Figure 4 for the association of LSMM with toxicity ≥ grade 3 were much more reliable because of the low level of heterogeneity with an I^2^ of 0%. This can be explained by the clear definition of grade 3 toxicity according to CTCAE. Therefore, the meta-analysis for the association between LSMM and toxicity ≥ grade 3 should be seen as more accurate and trustworthy compared to the meta-analysis for DLT.

A limitation of this study is the differences in measurement of SMM and diagnosis of LSMM. All included studies used CT which is the most commonly used and validated technique for SMM measurement [3,8,12], one study also included MRI measurements [29]. However, studies did measure SMM on different vertebrae levels. Most commonly L3 was used, which is also the most conventionally applied method in literature [12,13]. Several studies in this review used alternative vertebrae levels C3 or T4. The methods using these other vertebrae levels have been researched in recent publications but are less frequently used as L3 and some lack validation. The forest plot in Figure 4B shows studies that all used the same measurement technique although there was still a difference in the software used, as well as the time between scan and treatment start. This could influence the results, but it is difficult to estimate this influence as there is no previous research on these topics. Especially, the time between scan and treatment start is difficult to interpret as many studies do not report the used time frame. Future research should take this into account for their study design and the results they report.

Furthermore, the definition of LSMM varies between studies. Although some studies use SMI as a continuous variable, most determine a cut-off value to define the presence of LSMM. Most studies use cut-off values from previous publications in similar populations with larger sample sizes. Within this review, the most frequently used cut-offs were those determined by previous studies performed by Prado et al. [45] and Martin et al. [46]. Additional confusion in the already complex field of cut-off values is caused by the incorrect citation of these cut-off values. In this review, five studies cited the cut-off values of Prado et al. [45] but used cut-off values that deviate from those published in the original study. This variation in cut-off values could explain the large range in the occurrence of LSMM, which can be observed in literature as well as in this review (12.2–89.0%). For the optimal diagnosis of LSMM, a universal cut-off value would be preferable. This could be done in a large population of healthy individuals where two standard deviations below average SMI could be seen as a cut-off for LSMM.

The leading theory behind the association between LSMM and increased risk of toxicity relates to the influence of LSMM on drug distribution. Patients with LSMM have a decreased LBM, as muscle mass is a large contributor to LBM. This could cause increased drug levels in the plasma of patients with LSMM and thereby increase the risk of toxicity [6,8,9,11]. Many studies in this review consisted of populations treated with a combination of anti-cancer drugs using different dosing regimens, which makes it challenging to compare the drug distribution. Therefore, we specifically focused on studies focused on monotherapy. There was a trend showing increased OR with an increased volume of distribution. This can be seen in the forest plots, as sorafenib has a higher OR for toxicity occurrence when compared to cisplatin and carboplatin, and this correlates with the higher volume of distribution of sorafenib (Figure 5A,B). However, no definitive conclusions can be drawn yet since the sample size in these studies was too low.

Besides the distribution of anti-cancer drugs, many other treatment characteristics could be influenced by changes in SMM. To further investigate this, studies would be needed that observe similar populations treated with different anti-cancer drugs, preferably as monotherapy. However, this might be challenging to accomplish as many treatment regimens consist of combined anti-cancer drugs and the possible addition of radiotherapy or surgical procedures. This review showed a large variety of treatment details such as concomitant radiotherapy, treatment intent, and the possibility to use chemotherapy as an adjuvant treatment. There is previous research on the influence of treatment details such as the research by Ganju et al. [28], which showed that LSMM is associated with prolonged radiation breaks in head and neck cancer patients who underwent chemoradiotherapy. However, to fully investigate this association a meta-analysis should be focused on specifically chemoradiotherapy or adjuvant chemotherapy. This review is not designed to draw conclusions on those topics, and therefore, future research is needed. Another option is to further research the mechanism that causes this decrease of SMM and by that identify how LSMM influence adverse events.

The studies included in this review all investigated the association between pre-treatment LSMM and the occurrence of toxicity. Several studies also investigated the relationship between the change in SMM during treatment and increased toxicity. However, reverse causality could not be excluded from these observational studies. Randomized intervention studies are needed to elucidate whether diet, exercise, or supplements could reverse or prevent a decrease in SMM during systemic treatment and whether this leads to a lower risk of toxicity. Another strategy to produce better treatment outcomes is to adapt treatment regimens based on the presence or absence of LSMM, although this would require a universal cut-off value. Alternatively, the dosing of anti-cancer drugs could be adapted to be based on SMI as opposed to weight or body surface area. This would also require randomized trials to demonstrate the superiority of SMI dosing above current dosing methods.

## 5. Conclusions

Based on the association between LSMM and toxicity ≥ grade 3 according to the CTCAE, it can be concluded that the predictive value of LSMM for toxicity of anti-cancer drugs can be observed across cancer types and patient populations. This information increases the need for further research into interventions that could treat LSMM as well as the possibility to adapt treatment regimens based on the presence of LSMM. Additional research should also be done to validate measurement methods, create universal cut-off values, monitor changes in SMM during treatment, and investigate the influence of concurrent treatments.

## Figures and Tables

**Figure 1 jcm-09-03780-f001:**
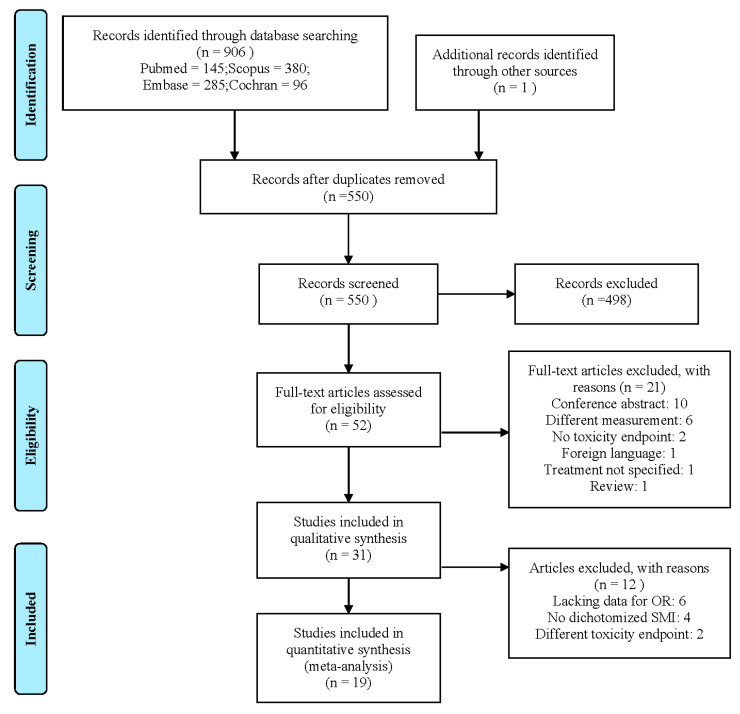
Preferred Reporting Items for Systematic reviews and Meta-analyses (PRISMA) flowchart detailing the study selection process.

**Figure 2 jcm-09-03780-f002:**
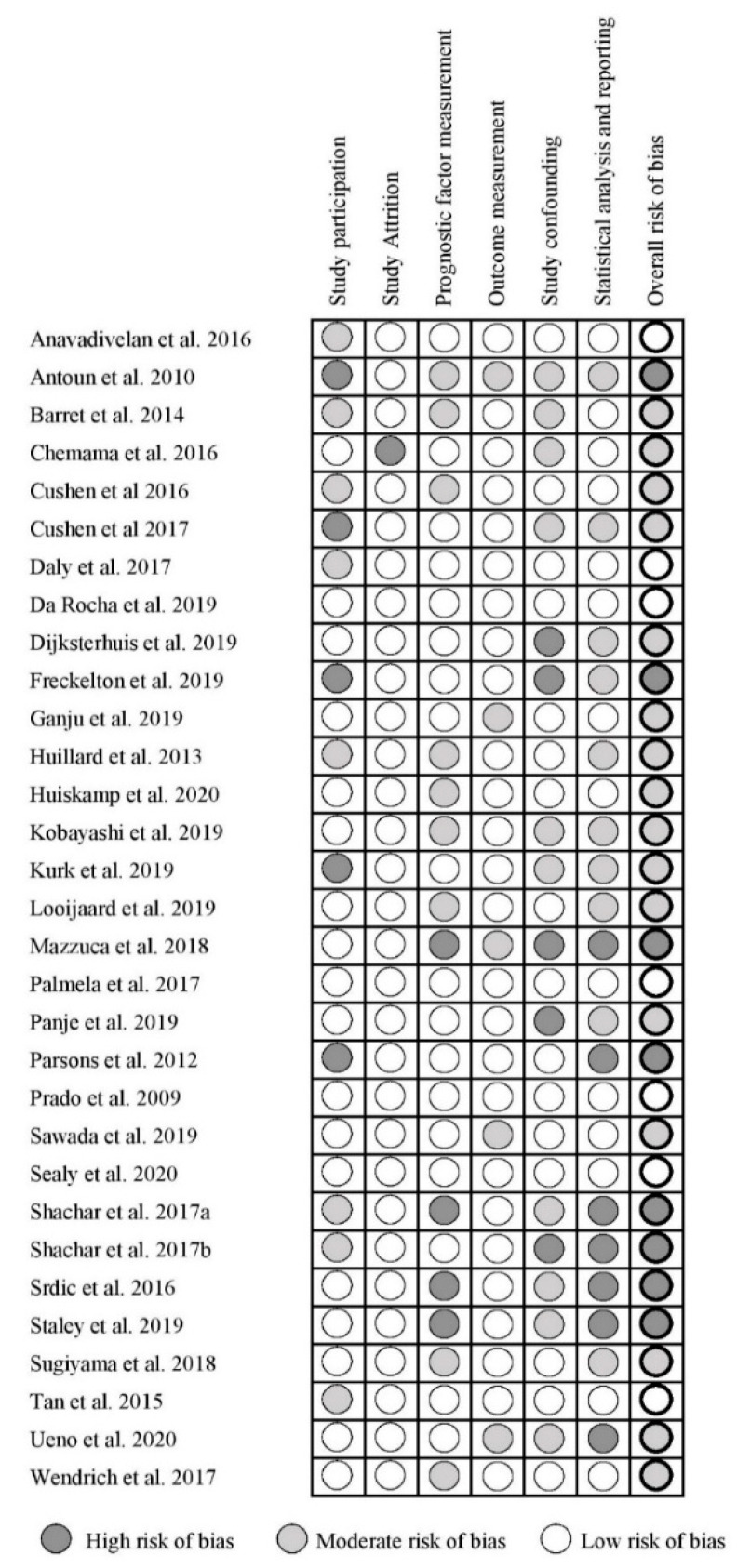
Quality in Prognostic Studies (QUIPS) for the included studies.

**Figure 3 jcm-09-03780-f003:**
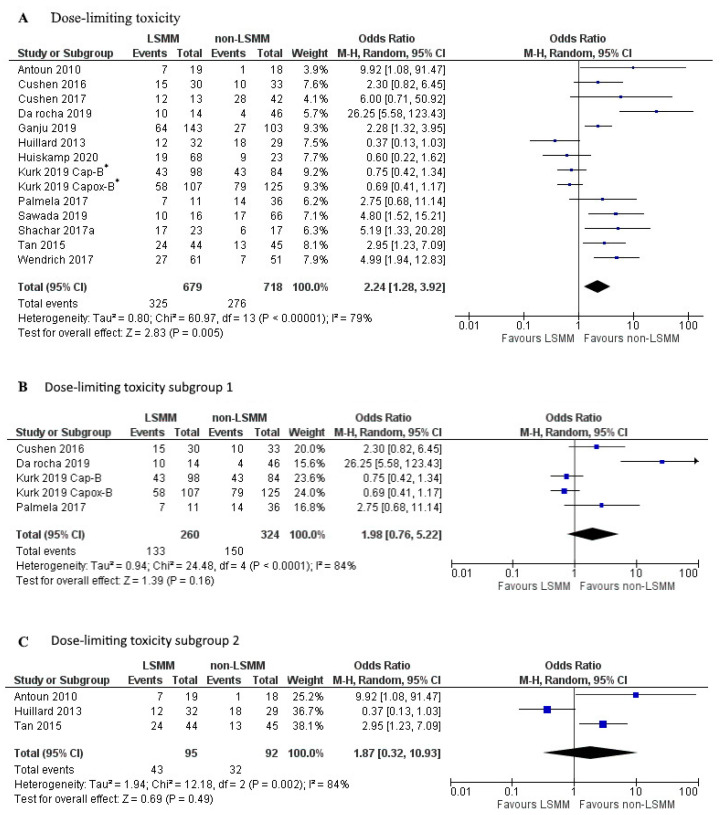
Forest plots for the association between low skeletal muscle mass (LSMM) and the odds to develop anti-cancer drug toxicity, specifically dose-limiting toxicity (DLT). (**A**) shows the odds to develop toxicity for all included studies with DLT as the toxicity endpoint. (**B**) shows the odds to develop DLT for a selected group of studies that besides the same toxicity endpoint also share the same cut-off value established by Martin et al., 2013 [46], as well as the same measurement technique using CT at the L3 vertebrae. (**C**) shows the odds to develop DLT for a second selected group of studies that share the same cut-off value established by Prado et al., 2008 [45], as well as the same measurement technique using CT at the L3 vertebrae. For each forest plot, the combined effect of the studies is plotted with a black diamond. * The patient population in the study by Kurk et al., 2019, received sequential treatments. The odds ratio was determined for each treatment separately and therefore entered separately into the forest plot.

**Figure 4 jcm-09-03780-f004:**
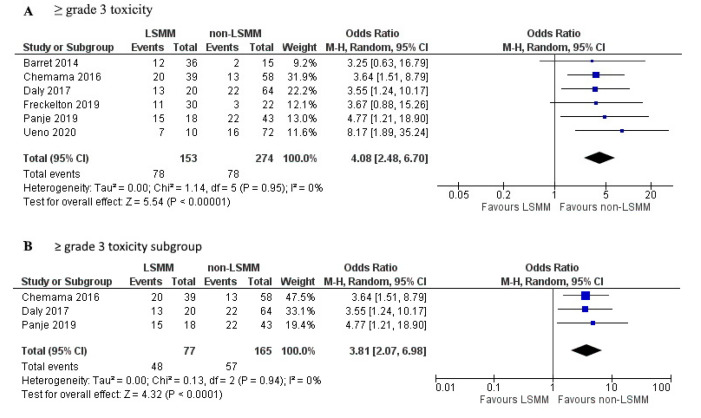
Forest plots for the association between low skeletal muscle mass (LSMM) and the odds to develop anti-cancer drug toxicity, specifically (**A**) toxicity ≥ grade 3 which was used as the toxicity endpoint in 6 studies. (**B**) shows a selection of studies that besides the same toxicity endpoint also used the same cut-off values established by Martin et al., 2013 [46], as well as the same measurement techniques using CT at the L3 vertebrae. For each forest plot, the combined effect of the studies is plotted with a black diamond.

**Figure 5 jcm-09-03780-f005:**
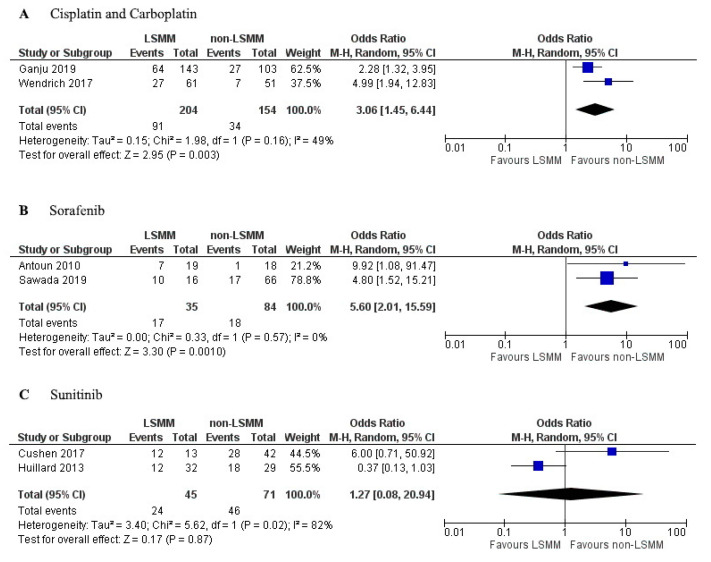
Forest plots for association between low skeletal muscle mass (LSMM) and toxicity specifically for monotherapies used in multiple studies. (**A**) includes patients treated with cisplatin or carboplatin as a monotherapy; (**B**) includes patients treated with sorafenib as monotherapy; (**C**) includes patients treated with sunitinib as monotherapy. For each forest plot, the combined effect of the studies is plotted with a black diamond.

**Table 1 jcm-09-03780-t001:** Characteristics of included studies.

Author and Date	(*n*)	Type of Cancer	Measure LSMM	Occurrence LSMM *n* (%) or Mean (SD)	Location Analysed	Anti-Cancer Drug	Measure of Toxicity	Occurrence Toxicity *n* (%)
Anavadivelan et al. 2016 [19]	72	Oesophageal	1	31 (43.0%)	CT-L3	Cisplatin + 5-FU	DLT ^a^	Not given
Antoun et al. 2010 [20]	55	Renal cell	1	30 (54.5%)	CT-L3	Sorafenib	DLT ^a^	12 (21.8%)
Barret et al. 2014 [5]	51	Metastatic colorectal	1	36 (70.6%)	CT-L3	FP with/without oxaliplatin oririnotecan with/without cetuximab	≥grade 3 toxicity	14 (27.5%)
Chemama et al. 2016 [21]	97	Peritoneal carcinomatosis and colorectal	2	39 (40.0%)	CT-L3	HIPEC oxaliplatin + irinotecan	≥grade 3 toxicity	33 (39.0%)
Cushen et al. 2016 [22]	63	Metastatic castrate resistant prostate	2	30 (47.6%)	CT-L3	Docetaxel-based	DLT ^a^	22 (34.9%)
Cushen et al. 2017 [23]	55	Clear cell renal cell	3	13 (23.6%)	CT-L3	Sunitinib	DLT ^a^	40 (73.0%)
Daly et al. 2017 [25]	84	Metastatic melanoma	2	20 (23.8%)	CT-L3	Ipilimumab	≥grade 3 toxicity	35 (41.7%)
Da Rocha et al. 2019 [24]	60	Gastrointestinal	2	14 (23.3%)	CT-L3	5-FU+ leucovorin, FOLFOX, or paclitaxel + carboplatin	DLT ^a^ during first cycle	14 (23.3%)
Dijksterhuis et al. 2019 [26]	88	Esophagogastric	2	43 (48.9%)	CT-L3	CAPOX	≥grade 3 toxicity during first cycle	32 (36.4%)
Freckelton et al. 2019 [27]	52	Metastatic pancreatic ductal adenocarcinoma	1	30 (57.7%)	CT-L3	Gemcitabine + nab-paclitaxel	≥grade 3 toxicity during first cycle	14 (27.0%)
Ganju et al. 2019 [28]	246	Head and neck cancer	2	143 (58.0%)	CT-C3	Cisplatin, cetuximab, orcarboplatin	DLT ^a^	91 (37.0%)
Huillard et al. 2013 [7]	61	Metastatic renal cell	1	32 (52.5%)	CT-L3	Sunitinib	DLT ^a^ during first cycle	18 (29.5%)
Huiskamp et al. 2020 [29]	91	Head and neck	≤45.2 cm^2^/m^2^	68 (74.7%)	CT-C3 MRI-C3	Cetuximab	DLT ^a^	28 (30.8%)
Kobayashi et al. 2019 [30]	23	Inoperable soft tissue sarcoma	<39 cm^2^/m^2^	11 (47.8%)	CT-L3	Eribulin	≥grade 3 toxicity	16 (69.6%)
Kurk et al. 2019 [31]	414	Metastatic colorectal	2	198 (47.8%)	CT-L3	CAPOX-B or CAP-B	DLT ^a^	130 (56.0%) 111 (61.0%) ^b^
Looijaard et al. 2019 [32]	53	Colon	Continuous SMI	46.3 (8.9)	CT-L3	Capecitabine, CAPOX, 5-FU+leucovorin, or FOLFOX	DLT ^a^	41 (77.4%)
Mazzuca et al. 2018 [33]	21	Stage 1–3 breast	≤38.5 cm^2^/m^2^	8 (38.1%)	CT-L3	A combination of 2–3: adriamycin, paclitaxel, docetaxel, epirubicin, trastuzumab, 5-FU, or cyclophosphamide	≥grade 3 toxicity	Not given
Palmela et al. 2017 [34]	47	Stomach or gastroesophageal junction	2	11 (23%)	CT-L3	A combination of 2–3: epirubicin, cisplatin, 5-FU, oxaliplatin, docetaxel, leucovorin, or capecitabine	DLT ^a^	21 (44.7%)
Panje et al. 2019 [35]	61	Locally advanced oesophageal	2	18 (29.5%)	CT-L3	Docetaxel + cisplatin with/without cetuximab	≥grade 3 toxicity	37 (60.7%)
Parsons et al. 2012 [36]	48	Liver metastasis	1	20 (42.0%)	CT-L3	HAI oxaliplatin + leucovorin + 5-FU + bevacizumab	≥grade 3 toxicity	Not given
Prado et al. 2009 [6]	55	Metastatic breast	1	14 (25.5%)	CT-L3	Capecitabine	≥grade 2 toxicity	15 (27.3%)
Sawada et al. 2019 [37]	82	Hepatocellular	4	16 (19.5%)	CT-L3	Sorafenib	DLT ^a^	27 (32.9%)
Sealy et al. 2020 [38]	213	Head and neck cancer	Continuous SMI	L3: 51.62 (10.16) T4: 65.53 (12.60)	CT-L3 or CT-T4	Cisplatin or carboplatin	DLT ^a^	61 (29.0%)
Shachar et al. 2017a [39]	40	Metastatic breast	≤41 cm^2^/m^2^	23 (58%)	CT-L3	Paclitaxel, docetaxel, or nab-paclitaxel combined with trastuzumab, pertuzumab, or bevacizumab	DLT ^a^	23 (58.0%)
Shachar et al. 2017b [40]	151	Early breast	Continuous SMI	44.72 (6.86)	CT-L3	Adraimycin + cyclophosphamide	≥grade 3 toxicity	50 (33.1%)
Srdic et al. 2016 [41]	100	Non-small cell lung	1	47 (47%)	CT-L3	Platinum based chemotherapy with gemcitabine, paclitaxel or etoposide	≥grade 2 toxicity during first cycle	57 (57.0%)
Staley et al. 2019 [42]	134	Epithelial ovarian	≤41 cm^2^/m^2^	73 (54.5%)	CT-L3	Platinum and taxane-based chemotherapy	Dose delay or reduction	51 (38.1%) 50 (37.3%)^c^
Sugiyama et al. 2018 [43]	118	Metastatic gastric	1	105 (89.0%)	CT-L3	FP with cisplatin or oxaliplatin	≥grade 3 toxicity	Not given
Tan et al. 2015 [16]	89	Oesophago-gastric	1	44 (49.4%)	CT-L3	Cisplatin + 5-FU or epirubicin + cisplatin + capecitabine	DLT ^a^	37 (41.6%)
Ueno et al. 2020 [44]	82	Breast	5	10 (12.2%)	CT-L3	Epirubicin + cyclophosphamide	≥grade 3 laboratory toxicity	23 (28.0%)
Wendrich et al. 2017 [8]	112	Squamous cell head and neck	≤43.2 cm^2^/m^2^	61 (54.5%)	CT- C3	Cisplatin or carboplatin	DLT ^a^	34 (30.4%)

5-FU: 5-Fluorouracil; BMI: body mass index; CAP-B: capecitabine and bevacizumab; CAPOX: Capecitabine and oxaliplatin; CAPOX-B: Capecitabine, oxaliplatin, and bevacizumab; C3: cervical vertebrae 3; CT: computed tomography; FOLFOX: oxaliplatin, leucovorin, 5-fluorouracil; FP: fluoropyrimidine; HAI: hepatic arterial infusion; HIPEC: hyperthermic intraperitoneal chemotherapy; L3: Lumbar vertebrae 3; LSMM: low skeletal muscle mass; MRI: magnetic resonance imaging; NS: not significant; SMI: skeletal muscle index (skeletal muscle area/height^2^); T4: thoracic vertebrae 4. ^a^. DLT (dose-limiting toxicity): toxicity leading to dose reduction, treatment delay, or discontinuation; ^b^. Occurrence of DLT for CAPOX-B and CAP-B respectively; c. Occurrence of dose delay and dose reduction respectively. Definitions of LSMM: 1. Prado et al. 2008 [45] <52.4 cm^2^/m^2^ for men and <38.5 cm^2^/m^2^ for women; 2. Martin et al. 2013 [46] <43 cm^2^/m^2^ for men if BMI ≤24.9 kg/m^2^ or <53 cm^2^/m^2^ for men if BMI >25kg/m^2^ and <41 cm^2^/m^2^ for women; 3. 25th percentile <44.8 cm^2^/m^2^ vs. 75th percentile >63.2 cm^2^/m^2;^ 4. Fujiwara et al. 2015 [47]: ≤36.2 cm^2^/m^2^ for men and ≤29.6 cm^2^/m^2^ for women; 5. Caan et al. 2018 [48] <40 cm^2^/m^2^.

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
