# Peer review of "The Predictive Value of Low Skeletal Muscle Mass Assessed on Cross-Sectional Imaging for Anti-Cancer Drug Toxicity: A Systematic Review and Meta-Analysis"

_jcm, 2020, doi:10.3390/jcm9113780_

Round 1

Reviewer 1 Report

The authors have submitted meta-analysis of the value of LSMM to predict toxicity of anti-cancer therapies and conclude that LSMM increases toxicity and is worth to be considered before deciding a therapeutic regimen

The authors also address the need of fix and/or prevent LSMM in patients to be treated for cancer

The review addresses a relevant clinical issue.    is comprehensive, informative and clear. 

I have no major concerns

Author Response

We thank reviewer 1 for the positive feedback.

Reviewer 2 Report

This is a properly designed systematic review and meta-analysis evaluating the predictive value of low skeletal muscle mass (LSMM) assessed on cross-sectional imaging for anti-cancer drug toxicity.

The study regularly meets the Preferred Reporting Items for Systematic Reviews and Meta-Analyses (PRISMA) criteria. Quality in Prognostic Studies (QUIPS) is also provided, allowing a proper evaluation of the risk of bias among the considered studies.

The work is well-conceived and conducted, though shows some limitations:

  • A wide range of LSMM (12.2–89.0%) and toxicity (21.8–77.45%) is reported among the selected studies. The former is related to the different cut-offs used to dichotomized SMM throughout the mentioned papers; the latter to the huge variability in definition of toxicity, especially dose-limiting one (DLT). This is obviously a point that could jeopardize the findings of the analysis.
  • The use of different cut-offs for LSMM definition is a significant and well-known limitation for comparison of series considering LSMM as predictor of bad outcome. The selected studies indeed used 5 different thresholds, posing a significant dilemma in terms of homogeneity of results. Since most of the studies employed Prado’s cut-offs (Prado et al., 2008), I think that the authors should particularly focus on these ones.
  • Only the results of association analysis between LSMM and toxicity >= 3 were linked to a low heterogeneity across studies (χ2 = 1.14 and I2 = 0%). As clearly stated, this is probably due to the clear definition of grade 3 toxicity - according to CTCAE - as opposed to other endpoints, especially dose-limiting toxicity (DLT). DLT was defined as toxicity leading to dose reduction (how much?), treatment delay (of what kind?) or discontinuation (with hospitalization or not?). This extreme variability in DLT definition makes the finding of the specific analysis quite questionable, not only theoretically, but also from a statistical point of view (χ2 = 60.97 and I2 = 79%; high heterogeneity across studies).
  • Different methods were used for SMM assessment through the selected studies. Some authors used only CT, others CT and MRI. In many works SMM evaluation was performed at L3 level, while in few others at C3 or T4. Moreover, from a theoretical point of view, there could be a role in influencing the clinical outcome (i.e. toxicity) for the heterogeneous “scan-to-treatment” time among the selected studies, which may determine significant differences in SMM definition. In few studies this time is not even detectable (N/A) and, in general, an evaluation of its role is lacking. I think that the authors should better specify their inclusion criteria by taking into consideration these points.
  • The potential confounding role of concurrent radiotherapy is not assessed at all, though it’s suggested in the manuscript.
  • QUIPS assessment underlines a potential high risk of bias for most of the selected studies; although a random-effects model was used, this finding might jeopardize the evidences of the meta-analysis.
  • Methods other than CT and MRI have also been recently validated for LSMM assessment, such as ultrasonography of larger muscle groups (e.g. rectus femoris) with evaluation of cross-sectional muscle area (Mueller et al, 2016; Galli et al, 2020). This should be mentioned in the article, since this technique has the potential advantage of standardizing the so-called “scan-to-treatment” time, one of the limitations previously mentioned.

While recognizing the difficulty in obtaining an adequate sample size for meta-analysis, I think that an effort should be made in selecting more homogeneous studies, especially in terms of modality of SMM assessment (CT), SMM cut-offs (maybe Prado et al., 2008), outcome definition (exclusively CTCAE toxicity >= 3) and potential confounders (concurrent RT), otherwise it will be difficult to draw meaningful conclusions from the present analysis. I know that most of the stated limitations are cited within the Discussion, but I think that they should also have an influence on the inclusion criteria of the study. Trying to address the afore-mentioned points.

Author Response

The concerns of reviewer 2 were addressed in the following manners. The variability in definition of toxicity as well as the different cut-offs for SMM might indeed influence the findings, as previously stated in our discussion. This point was further highlighted and elaborated in the discussion (line 320-327). Additionally, the cut-offs were mentioned as having an influence on the heterogeneity in the meta-analysis. This was addressed by performing additional analysis with subgroups that match in their cut-offs, as well as, the measurement technique (CT) and the vertebrae used (L3)(line 210-241, figure 2 and 3). This also addresses some of the other problems that the reviewer raised regarding the heterogeneity in the analysis. Subgroups should minimize the heterogeneity and the subgroups still demonstrate that DLT is not a reliable toxicity measure and that the analysis on CTCAE grade 3 toxicity is much more reliable. The reliability of these results was further taken into account in the discussion and the conclusion(line 320-327,line 392). We choose not to amend our inclusion criteria as to not limit the amount of studies/patients that we could analyse. However, additional analysis of subgroups allows us to create results that also show the results under more restricted conditions.

The point regarding the scan-to-treatment time was further addressed in the discussion. The lack of information reported in the publication made it difficult to incorporate this variable in our analysis. Therefore we added the advice that future research should report the time between scan and treatment start as well as take this into account in their study design and data analyses(line 336-338).

The reviewer mentions the possible confounding role of radiotherapy. This is indeed a possibility as well as the confounding role of surgical procedure performed before the start of chemotherapy. This review was not designed to answer these questions however, it was included in the discussion that future research could shed more light on these topics(line 371-377).

In the introduction we now have included some information on the other techniques for LSMM assessment besides CT and MRI(line 63-67).

Additionally, the conclusion was rewritten as that required improvement according to the reviewer(line 392-397).

Reviewer 3 Report

The prevalence of low skeletal muscle mass (LSSM) in patients with cancer is high. The association between LSMM and toxicity from anti-cancer treatment is frequently observed in clinical practice and has been the subject of investigation in several studies of varying sample sizes and cancer types. The present systematic review and meta-analysis aims to give a comprehensive overview of the literature and data regarding the predictive value of LSMM for anti-cancer drug toxicity in a mixed cancer population and estimate the overall effect in a meta-analysis. The authors searched several databases for relevant studies and used the PRISMA guidelines to narrow down the selection according to stated inclusion and exclusion criteria. The QUIPS tool was used to assess potential bias of the included studies. The authors are able to demonstrate a clear association between LSMM and increased toxicity from anti-cancer treatment, and they emphasize in their conclusion the importance of further research into treatment strategies for LSMM as well as looking into methods for adapting treatment according to LSMM.

I found this paper both interesting to read and relevant to clinical practice and further research. It reads well, and the background, methods, results and discussion are mostly presented clearly, and the limitations of the study are carefully discussed. However, I have some minor remarks:

In the methods section I think it should be stated how many reviewers independently performed the screening and selection, and, if more than one, how disagreements between reviewers were resolved. It should also be stated if and how records were identified outside the database search (typically through reading reference lists of already identified papers) as this is referred to in Figure 1.

In the results section in Figure 1 906 records are identified, while in the text (line 137) it says 907. Also, the number of additional records identified in Figure 1 is not stated. Please, clarify.

In line 188 it says >= grade 3. Should it be dose limiting toxicity (DLT)?

In line 196 it says 18 studies were included in the meta-analysis, while in Figure 1 19 are reported to be included. Please, clarify.

I was not able to identify many similar publications, however, there are some systematic reviews, with or without meta-analyses, considering toxicity and LSMM/sarcopenia, and the authors could consider comparing results with these to evaluate the validity of their own findings.

The authors discuss LSMM as a direct cause of increased toxicity in relation to distribution volumes and anti-cancer drug characteristics. While I find this interesting and relevant, I miss a broader discussion of indirect causes/confounders to the association between LSMM and toxicity. LSMM in patients with cancer is often associated with cachexia – a syndrome composed of weight loss/reduced muscle mass, anorexia and reduced food intake. It leads to increased morbidity, reduced physical function and increased mortality. Thus, there are several aspects of cachexia that potentially could contribute to increased toxicity, and I think a more in-depth discussion of this would emphasize the relevance of the paper’s findings.

Author Response

The concerns provided by reviewer 3 were addressed in the following manner. The study selection was performed by one researcher. This information was added to the methods section(line 111). The method for finding additional records outside of the search was added to the methods sections(line 107). This yielded one additional article. This information was added to the results section and thereby solves the discrepancy between the PRISMA chart and line 137(line 157-158).

Line 188 mentioned grade 3 toxicity, this should indeed by DLT(line 208). Line 196 reported that 18 studies were included in the meta-analysis, this should indeed be 19. This was fixed accordingly(line 263).

We were not able to identify any meta-analysis that we felt would have added value if we compared our results to theirs.

As mentioned by the reviewer, LSMM is indeed often associated with cachexia. There was an additional line added to the introduction regarding this topic(line 78-79). However, it is not included in the discussion as this review is unable to draw any conclusion or provide advice on that topic. The distribution volumes as they relate to LSMM were specifically discussed because it was an included topic in the analysis. Therefore, cachexia and other mechanisms that could cause LSMM were not included in a broader debate.

The limitations that this review and meta-analysis face have been addressed in the discussion and are taken into account when drawing the conclusions. The additional analyses which were performed provides a more in depth look at the heterogeneity of the study population.